# Design of a Circularly Polarized Micro-Strip Antenna for Aircraft Tracking Based on BeiDou III Compatible with Multi-Navigation System

**DOI:** 10.3390/mi14112083

**Published:** 2023-11-10

**Authors:** Zhenyang Ma, Xinyi Huang

**Affiliations:** 1Key Laboratory of Civil Aircraft Airworthiness Technology, Civil Aviation University of China, Tianjin 300300, China; 2Sino-European Institute of Aviation Engineering, Civil Aviation University of China, Tianjin 300300, China; xyhuang_1999@163.com

**Keywords:** micro-strip antenna, RHCP, aircraft tracking, meandering technique, defected ground structures (DGS)

## Abstract

This paper proposed a right-handed circularly polarized (RHCP) micro-strip antenna for multi-navigation system applications. The size of the antenna is 70 mm × 70 mm × 2 mm, which is fabricated on an FR4 substrate. A meandering technique on a patch layer and asymmetrical defected ground structures (DGS) are employed to achieve the purpose of miniaturization and increase the bandwidth of the axial ratio. The prototype of this antenna is fabricated according to simulations where the bandwidth of return loss, bandwidth of axial ratio, and radio pattern are further testified. The bandwidth of return loss (S_11_) and axial ratio (AR) of the antennas are from 1.540 GHz to 1.612 GHz and 1.554 GHz to 1.601 GHz, which would be available for L1 of GPS, L1 of SBAS, E1 of Galileo as well as B1I and B1C of BDS-3, the last two of which can be used for aircraft tracking. The relative bandwidth is 2.98%, which satisfies the standard of wide-band patch antennas.

## 1. Introduction

With the development of the civil aviation industry, aircraft are favored for efficiency in medium- and long-distance travel. Air safety is always the critical issue in transportation. With the increasingly dense air routes, which contain both passenger and cargo traffic, high-accuracy tracking and location of aircraft has become the focus of research [1,2,3]. The realization of this function relies on the navigation system and related airborne equipment, and the four major navigation systems in the world are as follows: Beidou Navigation Satellite System (BDS), Global Positioning System (GPS), Galileo satellite navigation system and Global navigation satellite system (GLONASS) [4,5,6,7].

An antenna is a critical component of airborne equipment. Within the corresponding operating frequency band of targeted navigation systems, an antenna serves as the transceiver of navigation information that would be applied for real-time positioning [8,9]. Since the accuracy of tracking is affected by antennas, it is necessary to design an antenna that can meet airworthiness standards. Taking BDS-3 as an example, the antenna in service is generally required to meet airworthiness regulation given by the document code CTSO-2C604a [10].

Up to now, three polarization modes of antennas have been proposed, namely linear polarization, elliptical polarization, and circular polarization. Recent research has shown that circularly polarized waves will benefit from their strong ability to resist multi-path interference and polarization mismatches affected by various environmental factors. They can also be converted into linear polarized waves as needed [11,12,13]. In this case, circular polarization becomes one necessary feature for navigation antennas. For receiving signals, the direction of circular polarization needs to be right handed [14].

In the design of navigation antennas at present, in addition to the requirements of wide operating frequency band and gain, it is also necessary to take the installation environment into account. For example, antennas work with other complex avionics, and it is significant to reduce the size to improve the portability [15,16]. Among all circularly polarized antennas, micro-strip antennas possess the advantages of small size, simple manufacturing process, and good consistency, making it feasible for airborne navigation antennas.

At present, the narrow bandwidth of both return loss (S_11_) and axial ratio (AR) is the limit of the micro-strip antenna [17,18]. In order to resolve this problem, some research has been conducted to optimize the structure of antennas, such as introducing capacitive disk feed structure [19], adding short circuit slight nails [20,21], or widening bandwidth by applying short circuit pins [22]. On this basis, Su et al. have proposed to expand the antenna bandwidth by using a Wilkinson power division network to adjust the length of the feed line [23]. Although such methods can effectively improve antennas’ performance, introducing additional structures makes the antenna structure more complex and affects the integration of the equipment.

Unlike the above methods, some efforts have revolved around stacking the radiation patches of the navigation antenna. Zhang et al. have proposed a circularly polarized antenna composed of four layers as metal surfaces with three layers as dielectric substrates, where the bandwidth was successfully expanded based on coupling between coherent layers [24,25]. Nevertheless, stacking multiple layers of patches may complicate the structure of the antenna and increase its volume, making the design of the feed more difficult. The coupling between layers and feed points will worsen the performance indicators of the antenna. Another possibility is to build a high-impedance surface on the antenna [26]. By optimizing the ground of the antenna, the polarization state of electromagnetic waves is theoretically controllable; thus, it would be practicable to expand bandwidth and increase the number of frequency points. This kind of method would be a potential candidate for BDS applications.

In this paper, the designed wideband RHCP micro-strip antenna is proposed, which would be available for aircraft tracking in B1I and B1C for BDS-3, L1 of GPS, L1 of SBAS as well as E1 of Galileo. In section two, the principles of the meandering technique and DGS are introduced, which provide the theoretical support for broadening the band width of S_11_ and AR. In section three, the antenna is designed in software HFSS (version 19.2). All the design indicators are satisfied after parameter optimizing. In section four, the antenna is tested by a vector network analyzer and microwave anechoic chamber. Comparisons of simulation and experimental results are given, which prove that the antenna meets the design requirements.

## 2. Theoretical Analysis

The resonant frequency of the antenna can be retrieved as given by Equation (1),
(1)$$f=\frac{c}{2L\sqrt{\varepsilon_{r}}}$$
where *f* denotes the resonant frequency, *c* is the speed of light, *L* refers to the length of the antenna patch, and *ε_r_* is the relative dielectric constant of the dielectric substrate.

From Equation (1), it can be observed that *f* will be affected by both dielectric constant and patch length. However, this will inevitably increase the risk of occurrence of both wave loss and the shortening of frequency bands. Therefore, our work will focus on structure optimization by applying meandering technology and defect ground structure to broaden the bandwidth of both the return loss and the axial ratio of antennas.

### 2.1. Meandering Technique

The meandering technique shown in Figure 1 was initially applied to the micro-strip antenna to reduce the size of the antenna by optimizing the effective surface current path [27]. In addition, by extending the current path of the original resonant mode, the resonant wavelength can be increased to reduce the resonant frequency of the patch.

### 2.2. Defected Ground Structure

Recently, more attention has been paid to DGS for performance enhancement of CP antennas. When the electrons move in the periodic potential field formed by the high-impedance surface, an electronic band structure will be generated due to the Bragg scattering [28], and the band gap energy between the bands will change the effective permittivity of the material, which plays an important role in suppressing surface waves. As a type of high-impedance surface, the defective ground structure can be used as the antenna floor to broaden the antenna’s bandwidth while maintaining the same size and structure of the antenna. The high impedance could be realized by means of the rectangle slots, which serve as a pass-band filter, as illustrated in Figure 2. The equivalent inductance *L_g_* and capacitors *C_g_* and *C_p_* in Figure 2 can be derived as shown in Equation (2),
(2)$$\left\{\begin{array}{l} C_{g}=\frac{B_{r}}{\omega_{2}\left(\frac{\omega_{1}}{\omega_{2}}-\frac{\omega_{2}}{\omega_{1}}\right)}\\ C_{p}=\frac{B_{p}}{\omega_{1}}\\ L_{g}=\frac{1}{\omega_{2}^{2}C_{g}} \end{array}\right.$$
where *ω*_1_ and *ω*_2_ refer to central frequency and cutoff frequency, and *B_r_* and *B_p_* are the electromagnetic induction intensity generated by *R_g_* and *R_p_*, respectively.

## 3. Antenna Design

### 3.1. Design Procedure

The main design steps of the antenna proposed in this article are as follows:Step 1: Determine the basic parameters of the proposed micro-strip antenna patches. The step will be carried out according to the Equations (4)–(8) given in Section 3.4.Step 2: According to the meandering technique, T-shaped and F-shaped gaps are added to adjust the current direction to generate the current that can flow counterclockwise and produce right-handed circularly polarized electromagnetic waves. At the same time, a square gap in the center of the patch is employed to increase the forward gain of the antenna. By means of the gap structure, the enhancement of the surface current path will then reduce the patch size of the antenna.Step 3: Design a defective structure by introducing two L-shaped slots on the diagonal of the same side of the coaxial feed, aiming at forming a high-impedance surface and expanding the axial bandwidth.Step 4: Scan the main parameters that would potentially affect the performance of antenna to determine the size of all antenna parameters.

### 3.2. Substrate Material

In the S-band and below, the FR4 material has the ability to act as an antenna dielectric substrate. To be precise, FR4 material is a glass fiber mixed with epoxy resin. The heat resistance of this material can reach 300 degrees, the operating frequency is about a few GHz, and the nominal value of the permittivity is about 4.4. FR4 material has nearly zero water absorption and can maintain its working performance in both dry and wet conditions, making it suitable for installation on aircraft. At the same time, its price is much lower than the dielectric substrate material of high-frequency boards, which meets the requirements of industry to reduce production costs.

At present, the nominal permittivity value of FR4 material is generally employed for antenna simulations as 4.4. However, due to the deviation of the dielectric substrate material selected by the manufacturer and the inevitable processing errors varying from 4.2 to 4.6, there will be significant differences between the simulated and measurement results of the antenna. This will lead to serious effects on the reliability of the antenna design. To reduce the impact of FR4 parameters on antenna design accuracy, we propose a feedforward optimization method that corrects the estimated value of FR4 through the deviation between sampled simulation and measured results. The critical correction steps are as follows:Step 1: Complete the preliminary design of the patch antenna using the nominal value of FR4 material;Step 2: Perform simulations to the designed antenna and extract corresponding S_11_ and AR distributions, respectively;Step 3: Measure the S_11_ and AR of the designed antenna using vector network analyzer;Step 4: Compare the simulation results of the antenna with the measured results to obtain the deviation distribution. Afterward, the values of FR4 in the simulation are adjusted within a certain range to obtain a series of simulated distributions under different values;Step 5: Perform least squares analysis on simulated and measured distributions. At this point, the dielectric constant with the smallest least squares estimation value is closest to the true value. In the design of process, permittivity *ε_r_* of FR4 is calculated and set as 4.52;Step 6: Perform antenna design and optimization with the permittivity of FR4 extracted from Step 5.

### 3.3. Antenna Patch Structure

In the recent implementations of micro-strip antenna, the features of patch could be generally summarized as circle and square one, respectively. Circular patch antennas possess more characteristics for structure design and more easily generate circularly polarized waves. In addition, circular patch antennas will generate higher radiation and lower losses. The radius of circle patch can be formulated given by Equation (3),
(3)$$\left\{\begin{array}{l} R=\frac{k}{{\left\{1+\frac{2h}{\pi \varepsilon_{r}k}[\ln(\frac{\pi k}{2h}+1.7726)]\right\}}^{\frac{1}{2}}}\\ k=\frac{8.794}{f\sqrt{\varepsilon_{r}}} \end{array}\right.$$
where *R* denotes the radius of patch, *f* is the center frequency of working band, *h* refers to the thickness of antenna, and *ε_r_* is the permittivity of substrate.

Based on this structure, Si et al. [29] have proposed a dual-circle micro-strip array antenna, where the designed antenna is able to work with lower ground coupling current and then decrease mutual coupling. In this design, the diameter of single circle patch measures 13.48 mm in the condition that *f*, *ε_r_*, and h equal 5.8 GHz, 4.4, and 2 mm, respectively. In the same condition, the length of square micro-strip antenna only measures 3.14 mm. This proves that square micro-strip antennas perform better in terms of miniaturization. Compared with circle patch antenna, the square micro-strip antennas now are proven as the reasonable choice in combination with other electronic components, especially in some complex environments such as airborne systems. Moreover, the square patches are easier to produce than the circle ones, saving production costs and resources. Therefore, a square structure will be employed in the design of our antenna, and corresponding simulations as well as analysis will be carried out to improve the effectiveness of our design.

### 3.4. Antenna Geometry

According to the analysis above, the circular polarized antenna serving the BDS-3 system is developed and designed with corresponding geometrical configurations as given in Figure 2. As shown in Figure 3a, a copper square patch with side length of *WT* is printed on the substrate (FR4, permittivity *ε_r_* = 4.52) of the antenna. According to [30], the length of the patch *WT* can be calculated according to the equations that follow.

The initial length of patch *L* can be preliminarily defined as Equation (4),
(4)$$L=\frac{c}{2f}{\left(\frac{\varepsilon_{r}+1}{2}\right)}^{-\frac{1}{2}}$$
where *c* refers to the velocity of light, *f* is the resonant frequency of antenna, and *ε_r_* is the permittivity of FR4. Height of the antenna *h* is set as 2 mm. Then, the effective permittivity of antenna could be defined as Equation (5),
(5)$$\varepsilon_{ref}=\frac{\varepsilon_{r}+1}{2}+\frac{\varepsilon_{r}-1}{2}{\left[1+12\frac{h}{L}\right]}^{-\frac{1}{2}}$$
where *ε_ref_* denotes the effective permittivity. Here, we introduce the length extension of patch given by Equation (6),
(6)$$\Delta L=0.412\frac{\left(\varepsilon_{ref}+0.3\right)\left(\frac{L}{h}+0.264\right)}{\left(\varepsilon_{ref}-0.258\right)\left(\frac{L}{h}+0.8\right)}$$
where ∆*L* refers to length extension. From Equation (4), we can obtain effective length as shown in Equation (7),
(7)$$L_{ef}=\frac{c}{2f\sqrt{\varepsilon_{ref}}}$$
where *L_ef_* refers to the effective length. The actual length *WT* could be calculated as Equation (8),
(8)$$WT=L_{ef}-2\Delta L$$

In this work, *WT* is calculated as 45.5 mm according to Equation (8).

On the patch of the antenna, a narrow slot with dimensions of *p* × *q* is embedded along the *x* axis. In this patch, a single probe feed is implemented at (*x*_0_, *y*_0_) and works for exciting right-handed CP radiation. By using the theory of the meandering technique, one T-shaped structure is added to each edge of the square patch, as given by Figure 3a. The root along each axis exhibits identical configurations, where the length of T-shaped roots is set as da. Moreover, F slots are deployed along the long edge of each F structure. The width of F-slots is set as St. Figure 4 shows the comparison of bandwidth of return loss before and after grooving.

As shown in Figure 4, after adding slots, resonant frequencies shift entirely to the left. Band width of the middle one is expanded at the same time. Figure 5 is the vector of current on the surface of the antenna that shows current moving exactly along the slots.

As shown in Figure 5, the direction of vector of current changes counterclockwise with the phase. It illustrates that the current rotates counterclockwise to produce right-handed polarized electromagnetic waves.

In order to improve antenna performance, especially AR, DGS is designed on the ground of the antenna. Figure 3b gives the DGS designed in this paper, which is composed of one rectangular-shaped slot in the ground’s center and two centro-symmetrical L-shaped slots. Figure 6 shows the comparison of axial ratio (AR) of the antenna before and after adding DGS. From Figure 6, it is clearly observed that a decline of AR occurs after adding L-shaped slots. Moreover, the center slot can be applied to shift the AR to the left slightly. These simulations have revealed that DGS is a method to achieve the purpose of expanding bandwidth of AR. After parameters optimization in Section 3.2, the AR on the working band is totally below 3 dB.

### 3.5. Parameters Optimization

The geometry of the antenna is further optimized based on the analysis of simulated bandwidth retrieved from software HFSS. As given by Figure 7, Figure 8, Figure 9 and Figure 10, it is noticed that both S_11_ and AR can be adjusted by changing parameter values.

Figure 7 shows that both S_11_ and AR shift to a high-frequency band with the increase in parameter *dw*. Figure 8 shows that S_11_ and AR move down with the increase in *db*. In Figure 9a, S_11_ moves up with increase in *p*, while in Figure 9b, 3 dB band of AR moves to the low-frequency band with the increase in *pg*. In Figure 10, as *X*1*g* increases, the axis ratio moves downwards to the high frequency, and the effect on S_11_ is not significant within the operating frequency band.

It is practical to adjust S_11_ and AR as a whole by changing parameters *dw* and *db*. Parameter *p* can be used for changing S_11_ separately, while *pg* and *X*1*g* are for AR.

Figure 11, Figure 12, Figure 13 and Figure 14 display the optimized performance of the antenna in simulations. Figure 11 gives the bandwidth of return loss of antenna below −10 dB, which is located from 1.53 GHz to 1.62 GHz. Figure 12 reveals that the bandwidth of axial ratio below 3 dB is in a range from 1.555 GHz to 1.597 GHz. Figure 13 and Figure 14 give the radiation patterns and impedance of antenna, respectively. It can be observed that our antenna is right-handed circular polarization. The geometrical parameters of antenna are summarized and listed in Table 1.

## 4. Experiments and Discussions

The antenna after validation is modeled and processed based on PCB printing. Figure 15a shows the layout of antenna in both top views and bottom views. To solidify our investigations, it is tested in a full anechoic chamber, shown in Figure 15c. The antenna was attached to the pedestal and secured with foam. All the parameters are tested by multi-probe spherical near-field test system in combination with a model vector network analyzer n5232B. For convenience, the comparison between the measured and the simulated result of S_11_ and AR are given in Figure 16 and Figure 17, where merely a small bandwidth deviation can be observed. Figure 17 is tested gain of the antenna. The results display good matching, with some slight deviations due to the fabrication tolerance. And all the parameters satisfy the antenna design indicators. The actual band of S_11_ and AR are from 1.540 GHz to 1.612 GHz and 1.554 GHz to 1.601 GHz, which indicate the working band of the design is the second one.

Figure 18 shows the measured radiation patterns, and Figure 19 gives the comparison between the measured radiation patterns and the simulated ones. It is noticed that the antenna is right-handed circularly polarized, the forward radiation gain is 1.39 dBic, and the ones at ±90 degrees are −7.8 dBic and −7.68 dBic, respectively. Although there exist some deviations between the measured one and simulated one, they are acceptable according to the previous research. The reasons forsuch imperfections can be summarized as follows:(1)Since the dielectric substrate material is FR4, its dielectric constant is 4.2 to 4.6, the dielectric constant of FR4 material used by the manufacturer cannot be guaranteed to be identical with the set value of 4.52 in thesofware HFSS (version 19.2). And the dielectric constant of the substrate material has a large effect on the resonant frequency of the antenna, which causes the difference between the simulation and the actual measurement.(2)In the manufacture of the antenna, high manufacturing accuracy is required due to the use of a loaded seam and defective ground structure. Manufacturing errors within the process allowable errors can also cause errors in the test results of the antenna’s performance.(3)When welding the coaxial feed probe, the corner of the L-shaped slot of the defective ground structure is inevitably blocked by the SM joint, resulting in a deviation in the measured results.(4)The unavoidable presence of non-insulating materials in the measurement environment would impact the measurement reliability.

According to the bandwidths of S_11_ and AR retrieved from experiments, the operating band is 0.047 GHz. The relative band width (RBW) is defined by Equation (9),
(9)$$\mathrm{RBW}=\frac{2(f_{H}-f_{L})}{f_{H}+f_{L}}\times 100\%$$
where *f_H_* and *f_L_* are the are the highest and lowest values of the operating band.

RBW of this design is calculated as 2.98%, which is greater than 1% and satisfies the definition of a wideband patch antenna. The following table shows a comparison between the proposed antenna and other designs.

Although there exist some deviations between the measurement results and the theoretical values, our proposed concept for micro-strip antenna design still has significant advantages compared to the existing methods in refs [31,32,33,34,35]. Table 2 provides a comparison of the measurement performance between other design methods and the proposed antenna. In the design of these antennas, FR4 were all deployed as substrates. Compared with other antennas, although the proposed one might not be an advantage in size, it shows an absolute superiority in AR and relative bandwidth. The relative bandwidth of the proposed antenna is 2.98%, while the ones for the rest are all less than 2%. It is verified that the proposed antenna can receive signals over a wider frequency band to provide more services for aircraft navigation.

## 5. Conclusions

In this work, a circularly polarized antenna is proposed for BDS-3 applications. By means of the meandering technique, the geometry of the antenna is obtained as 7 mm × 7 mm × 2 mm. Afterward, rectangular-shaped and L-shaped DGS are introduced to broaden the bandwidth of the axial ratio of the antenna. After a series of optimizations, the bandwidth of return loss and axial ratio of antenna can be retrieved as 1.540 GHz to 1.612 GHz and 1.554 GHz to 1.601 GHz. The suggested antenna is fabricated and experiments are carried out on analysis of S_11_ and radiation patterns. The performance of the antenna and quality of our work are validated according to the comparisons between simulated and measured results. The relative bandwidth of the antenna is determined as 2.98%. The suggested antenna also possesses high application potential compatible with L1 of GPS, L1 of SBAS, and E1 of Galileo.

## Figures and Tables

**Figure 1 micromachines-14-02083-f001:**
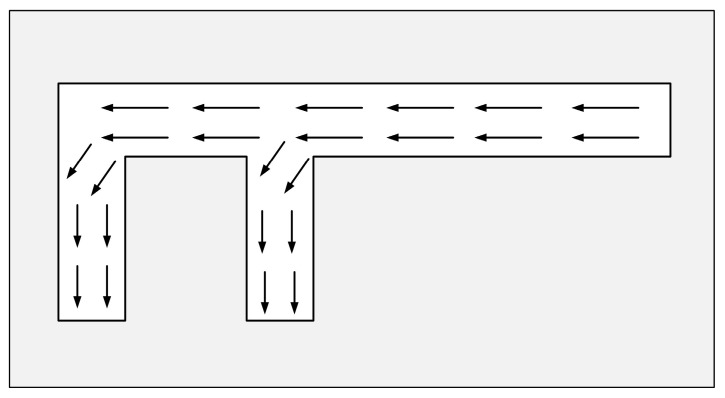
Schematic of meandering technique.

**Figure 2 micromachines-14-02083-f002:**
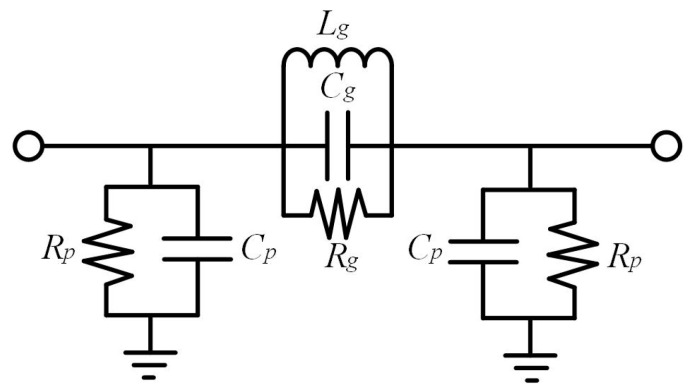
Equivalent circuit of DGS.

**Figure 3 micromachines-14-02083-f003:**
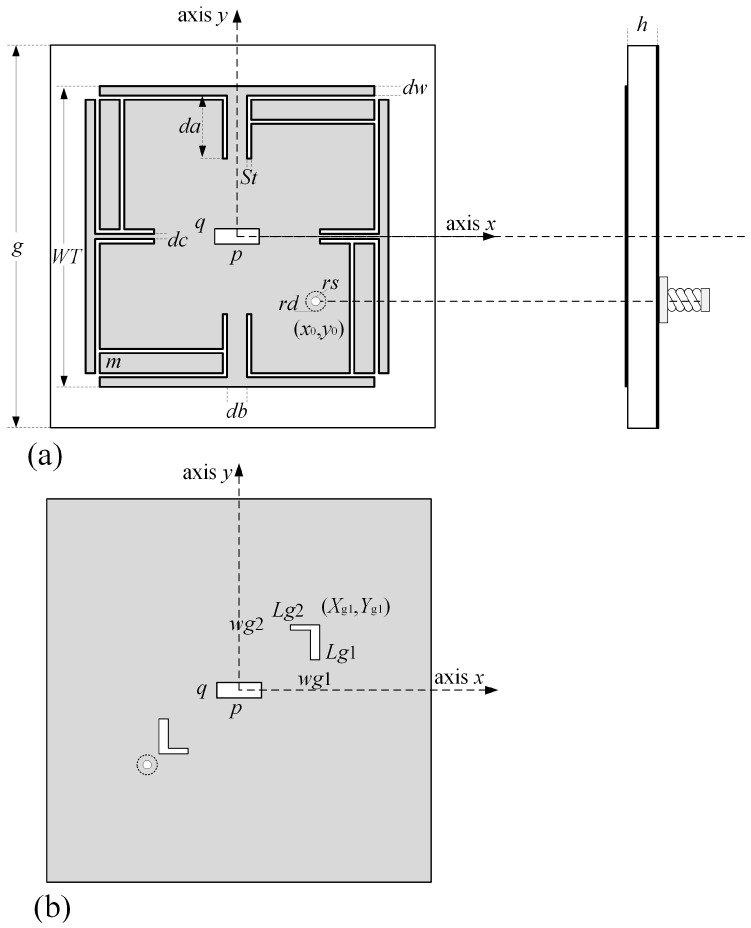
Geometry of designed antenna. (**a**) Top view of antenna. (**b**) Bottom view of antenna.

**Figure 4 micromachines-14-02083-f004:**
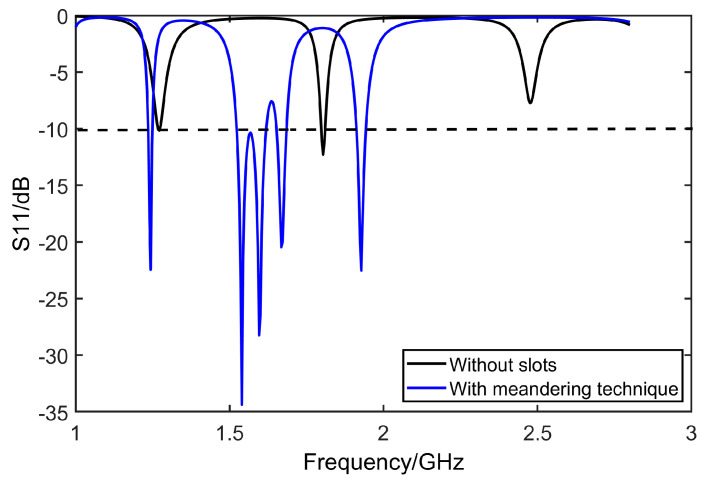
Comparison of S_11_ before and after using meandering technique.

**Figure 5 micromachines-14-02083-f005:**
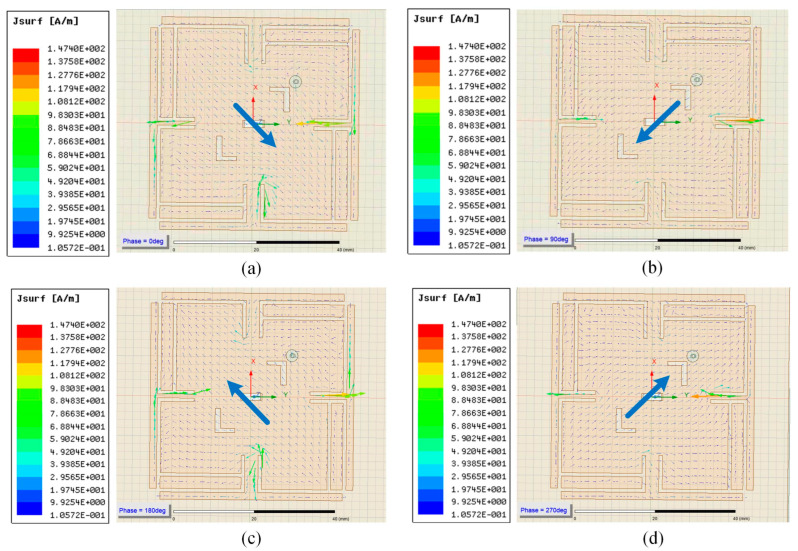
Vectors of current under different phases. (**a**) 0 deg; (**b**) 90 deg; (**c**) 180 deg; (**d**) 270 deg.

**Figure 6 micromachines-14-02083-f006:**
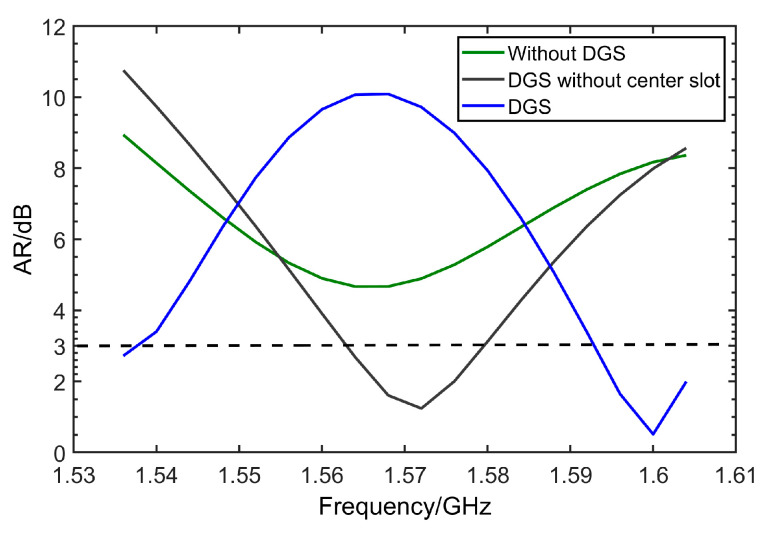
Comparison of AR before and after adding DGS.

**Figure 7 micromachines-14-02083-f007:**
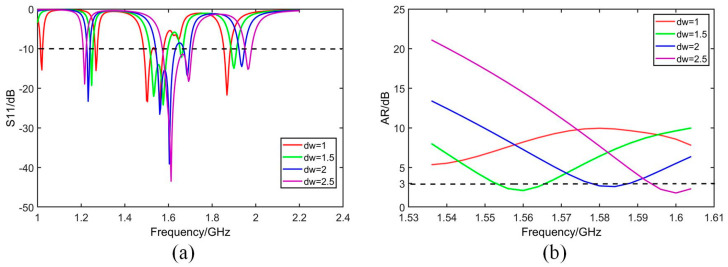
Simulated S_11_ and AR with different *dw*. (**a**) S_11_. (**b**) AR.

**Figure 8 micromachines-14-02083-f008:**
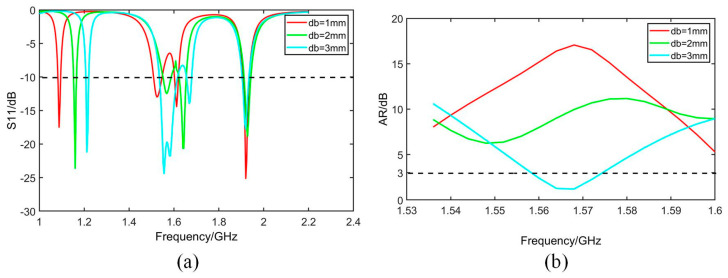
Simulated S_11_ and AR with different *db*. (**a**) S_11_. (**b**) AR.

**Figure 9 micromachines-14-02083-f009:**
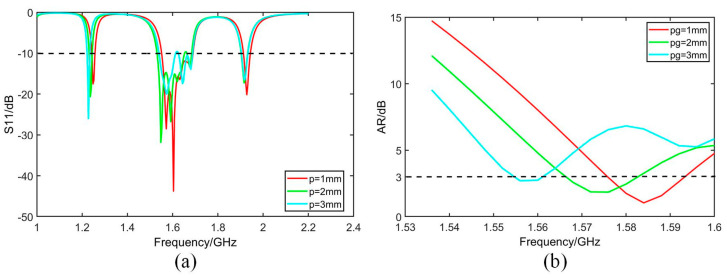
Simulated S_11_ and AR with different *p* and *pg*. (**a**) S_11_. (**b**) AR.

**Figure 10 micromachines-14-02083-f010:**
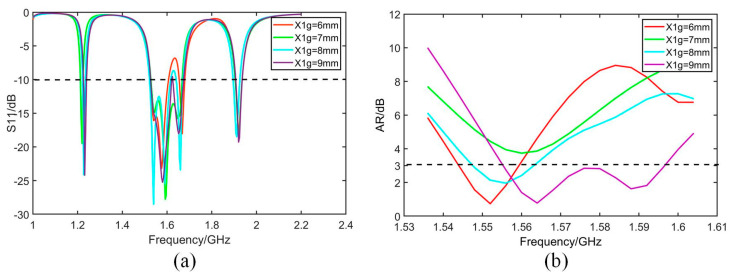
Simulated S_11_ and AR with different *X*1*g*. (**a**) S_11_. (**b**) AR.

**Figure 11 micromachines-14-02083-f011:**
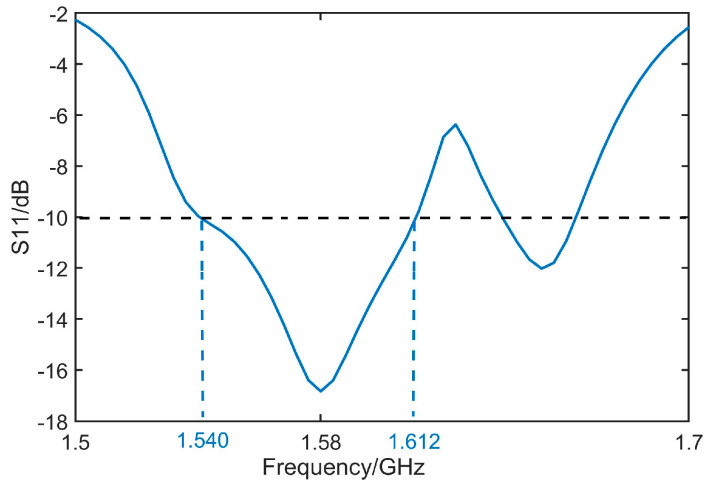
Optimized S_11_ of antenna.

**Figure 12 micromachines-14-02083-f012:**
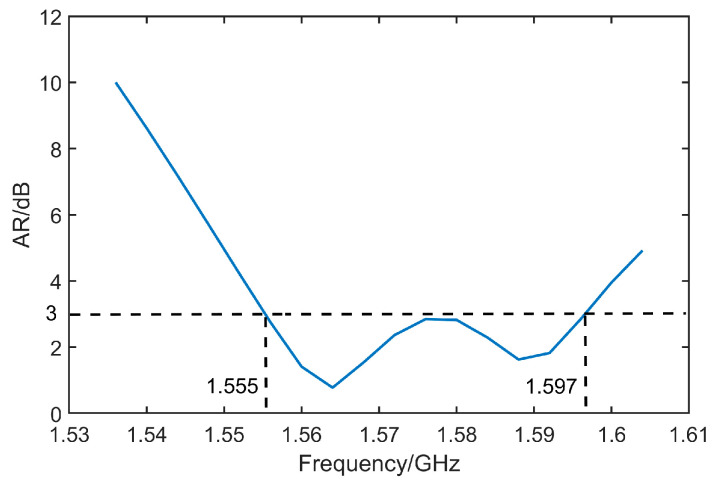
Optimized AR of antenna.

**Figure 13 micromachines-14-02083-f013:**
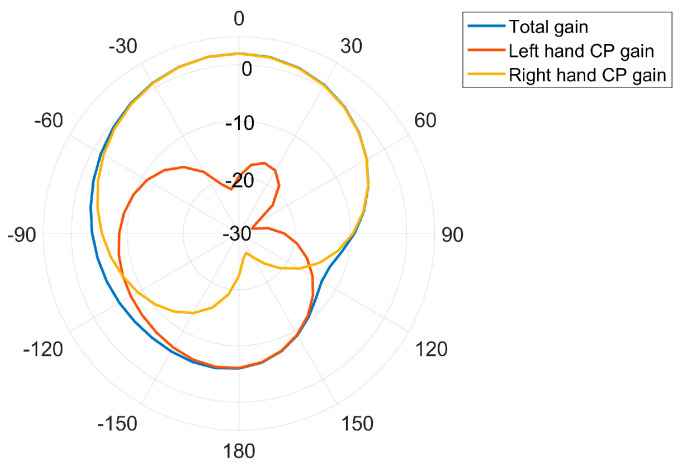
Radiation patterns of antenna.

**Figure 14 micromachines-14-02083-f014:**
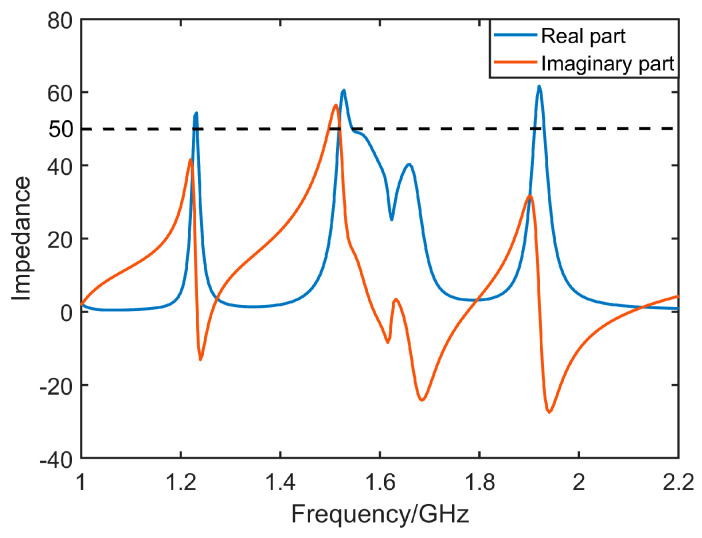
Optimized impedance of antenna.

**Figure 15 micromachines-14-02083-f015:**
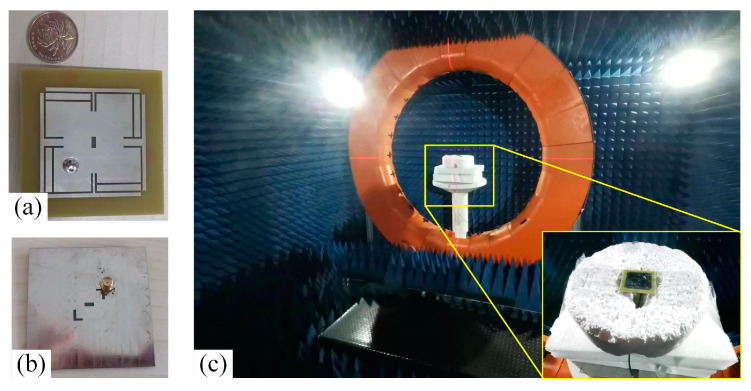
Measurement of S_11_ of antenna. (**a**) Top view of antenna. (**b**) Bottom view of antenna. (**c**) The proposed antenna tested in the microwave anechoic chamber.

**Figure 16 micromachines-14-02083-f016:**
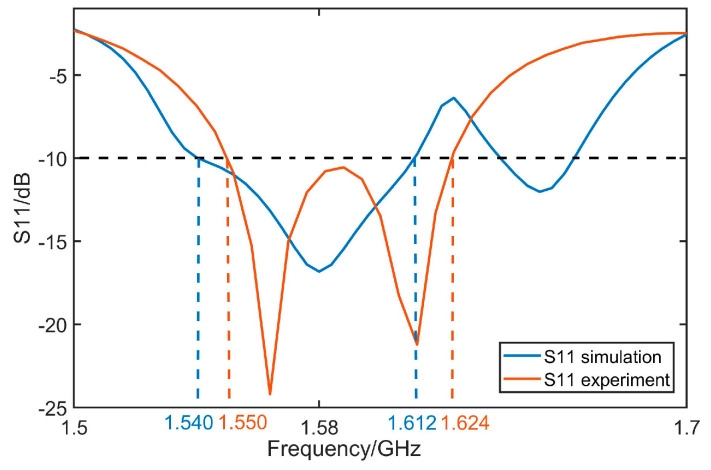
Comparisons of S_11_ between measured one and simulated one.

**Figure 17 micromachines-14-02083-f017:**
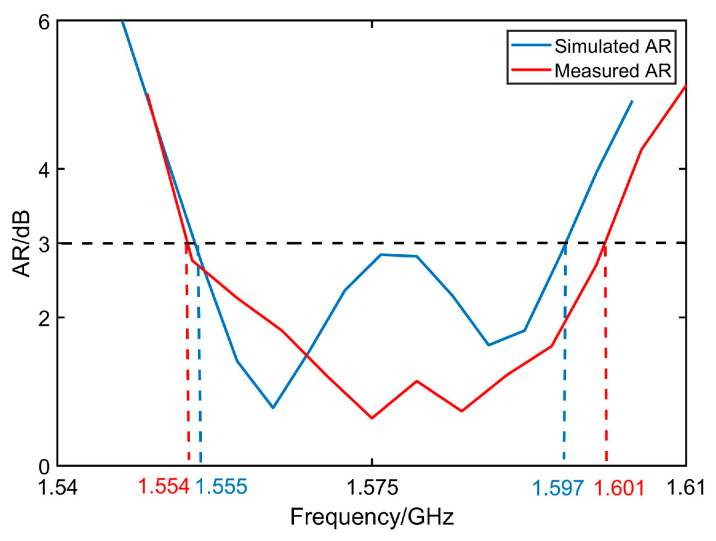
Comparisons of AR between measured one and simulated one.

**Figure 18 micromachines-14-02083-f018:**
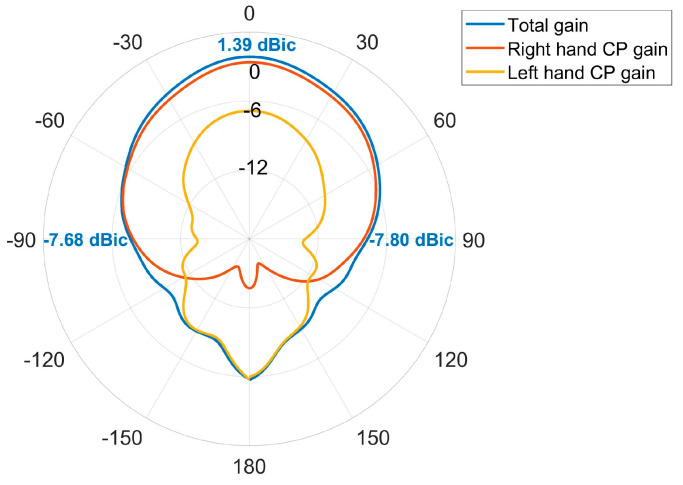
Measured radiation patterns of antenna.

**Figure 19 micromachines-14-02083-f019:**
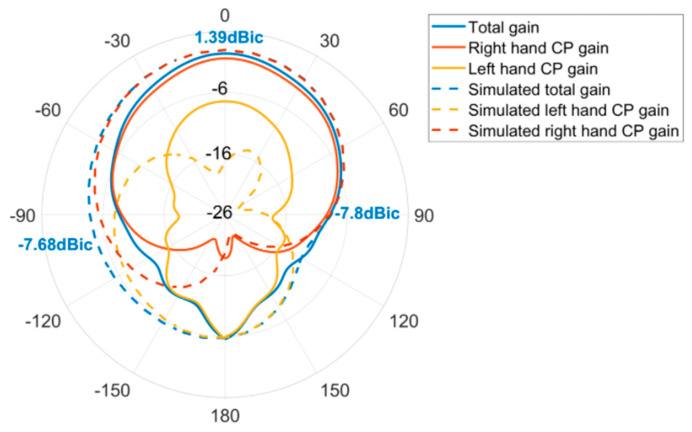
Comparison between measured radiation patterns and simulated ones.

**Table 1 micromachines-14-02083-t001:** Geometrical parameters as well as values of antenna.

Index	Variable	Value/mm	Index	Variable	Value/mm
1	*WT*	51.8	13	*x* _0_	10.4
2	*g*	70	14	*y* _0_	−10.4
3	*h*	2	15	*Xg*1	9
4	*da*	9.38	16	*Yg*1	9
5	*db*	3.4	17	*qg*	5
6	*dc*	1	18	*pg*	1.8
7	*m*	3.9	19	*Lg*1	6
8	*St*	0.9	20	*Wg*1	1.4
9	*dw*	1.8	21	*Lg*2	5
10	*p*	1.8	22	*Wg*2	1
11	*q*	5	23	*rd*	1.5
12	*rs*	0.5			

**Table 2 micromachines-14-02083-t002:** Performance comparison between the proposed antenna and other designs.

Designed Antenna	Size of Antenna	Center Resonant Frequency	Bandwidth of S_11_	Bandwidth of AR	Relative Bandwidth
Reference [31]	62 mm × 66 mm × 3.2 mm	1575 MHz	11.8 MHz	9.135 MHz	0.58%
Reference [32]	100 mm × 150 mm × 4 mm	920 MHz	50 MHz	15 MHz	1.63%
Reference [33]	60 mm × 60 mm × 3 mm	2250 MHz	120 MHz	26 MHz	1.15%
Reference [34]	90 mm × 90 mm × 1.6 mm	907 MHz	18 MHz	6 MHz	0.66%
Reference [35]	60 mm × 60 mm × 3.1 mm	2380 MHz	137 MHz	40 MHz	1.68%
Proposed antenna	70 mm × 70 mm × 2 mm	1575 MHz	72 MHz	47 MHz	2.98%

## Data Availability

The data will be made available upon a reasonable request to the corresponding author.

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
