# Peer review of "Design of a Circularly Polarized Micro-Strip Antenna for Aircraft Tracking Based on BeiDou III Compatible with Multi-Navigation System"

_micromachines, 2023, doi:10.3390/mi14112083_

Round 1

Reviewer 1 Report

Comments and Suggestions for Authors

This work proposed a RHCP micro-strip antenna for multi-navigation system applications. The theoretical analysis as well as the parameter study are given  in detail. I have some comments.

1) Can the authors give the reasons for the difference between the simulated and measured results.

2) It is better to give the design procedure to guide the design.

3) The comparison table is suggested to be added to highlight the advantages of the design.

Comments on the Quality of English Language

N.A.

Reviewer 2 Report

Comments and Suggestions for Authors

This paper deals with the design of a planar antenna with a circular polarisation. The antenna is mounted on an FR4 substrate. The fabrication is done and measurement was compared to simulation. Firstly, author has to explain the reason for the choice of FR4 substrate which is not suitable for low losses. In order to detail more this paper, author has to explain why he or she has a difference between simulation and measurement results for S11 and in the same time author has to  plot simulated and measured radiation pattern and to give more details about the obtained difference between the both results.

Comments on the Quality of English Language

The english of the paper can be more improved.

Reviewer 3 Report

Comments and Suggestions for Authors

The article concerns a circularly polarized moiscrostrip antenna intended for use in aircraft. It is written in a clear way. Below are my suggestions and questions regarding the article:

1. FR4 dielectric is popular, although it is characterized by a relatively large loss angle. what value was assumed in the calculations?

2. A broader explanation of the selection of slots and square patches would be appreciated.

3.What simulation environment was used for calculations?

4. What calculation method was used in the simulation software?

5. It should be written S11 with 11 as the subscript.

6. What optimization method was used?

7. It would be good to include a scale in Figure 5 so that the reader can understand what current value occurs in each place of the antenna. The direction of the current is very important, but so is the value.

8. It is a good idea to place a photo of the antenna next to a popular coin to illustrate its size. This is not a requirement, but a tip for the future.

Round 2

Reviewer 1 Report

Comments and Suggestions for Authors

I have no comments.